# Irisin Role in Chondrocyte 3D Culture Differentiation and Its Possible Applications

**DOI:** 10.3390/pharmaceutics15020585

**Published:** 2023-02-09

**Authors:** Francesca Posa, Roberta Zerlotin, Anastasia Ariano, Michele Di Cosola, Graziana Colaianni, Aldo Di Fazio, Silvia Colucci, Maria Grano, Giorgio Mori

**Affiliations:** 1Department of Clinical and Experimental Medicine, University of Foggia, Viale Pinto 1, 71122 Foggia, Italy; 2Department of Precision and Regenerative Medicine and Ionian Area (DiMePRe-J), University of Bari Aldo Moro, Piazza Giulio Cesare 11, 70124 Bari, Italy; 3Regional Complex Intercompany Institute of Legal Medicine, San Carlo Hospital, 85100 Potenza, Italy; 4Department of Translational Biomedicine and Neuroscience (DiBraiN), University of Bari Aldo Moro, Piazza Giulio Cesare 11, 70124 Bari, Italy

**Keywords:** cartilage regeneration, irisin, human articular chondrocytes (HACs), 3D pellet, chondrogenesis

## Abstract

Irisin is a recently discovered cytokine, better known as an exercise-induced myokine, produced primarily in skeletal muscle tissue as a response to exercise. Although the skeleton was initially identified as the main target of Irisin, its action is also proving effective in many other tissues. Physical activity determines a series of beneficial effects on health, including the possibility of counteracting the damage that is caused by arthritis to the cartilage of people suffering from osteoarthritis. Nevertheless, up to now, the studies that have taken into consideration the possible involvement of Irisin on the well-being of cartilage tissue are particularly limited. In this study, we postulated that the protective effect of physical activity on cartilage tissue may depend on the paracrine action of Irisin secreted during exercise; therefore, we analyzed the effects of Irisin, in vitro, on chondrogenic differentiation. To achieve this goal, three-dimensional cultures of commercially available human articular chondrocytes (HACs) were treated with the molecule under study. Our results revealed new crosstalk mechanisms between muscle and cartilage tissue. Furthermore, the confirmation of Irisin ability to induce chondrogenic differentiation could favor the development of exercise-mimetic drugs, with application relevance for patients who cannot perform physical activity.

## 1. Introduction

Bone and muscle are two tissues that influence each other, and it is known that exercise can prevent diseases related to them. Irisin, unknown for a long time, is the molecular mediator that couples the beneficial effects on bone and muscle. This molecule, produced by skeletal muscle after physical activity, is released into the circulation as a product of the membrane protein FNDC5 (fibronectin type III domain-containing protein-5) [1] that plays a key role in bone metabolism. In healthy mice, the intermittent administration of Irisin induces the formation of new bone and makes the skeleton more resistant to torsion [2]. In a mouse model of disuse-induced osteo-sarcopenia, the administration of Irisin is effective both on bone, where it prevents the reduction of bone mass, and on muscle, preserving its mass and increasing mitochondrial biogenesis [3]. The direct target bone cells of Irisin are osteoblasts [2] and osteocytes [4] on which it activates the phosphorylation of extracellular signal-regulated protein kinase (Erk1/2) and increases the expression of activating transcription factor 4 (Atf4), a fundamental transcriptional factor for cell proliferation, differentiation, and survival. In osteocytes, Irisin decreases the expression of Sclerostin, a potent physiological inhibitor of bone formation, both in vitro [4] and in vivo [3]. In addition, we showed that Irisin preserves osteoblast differentiation and activity in microgravity [5].

Recently, we have also described Irisin effect on the differentiation of mesenchymal stem cells from dental bud: dental bud stem cells (DBSCs) [6]. Irisin, in this cellular model, induces the expression of osteocalcin (OCN), a late marker of osteoblastic differentiation and causes greater deposition of the mineral matrix. These results indicate an important involvement of Irisin in the osteogenic differentiation process of DBSCs. Furthermore, considering the in vivo studies related to the effects of Irisin on the skeleton, this myokine could improve bone tissue metabolism in regenerative medicine procedures with mesenchymal stem cells.

In support of Irisin importance in the human musculoskeletal system, several observational studies have shown that the levels of circulating Irisin correlate with bone health [7,8,9] and lower Irisin levels were detected in a population of older adult subjects with mild/severe osteoporosis compared to non-osteoporotic controls [10].

Another degenerative pathology of the skeleton is osteoarthritis, characterized by the thinning of the articular cartilage. It is a degenerative joint disease that affects more than 10% of adults over the age of 60, characterized by pain and increased joint stiffness that often lead to disability, with a consequent negative impact on the overall functionality and quality of life of patients, as well as on healthcare costs [11].

Osteoarthritis and osteoporosis are bone disorders that usually affect people over the age of 50 and are associated with substantial morbidity and disability. Although these pathologies mainly affect the elderly, until recently the attention was focused on an alleged inverse relationship between the two disorders [12], which rarely coexist in a single person [13]. Recent advances in osteoimmunology have clarified that increased bone loss occurs not only in osteoporosis, but also in the early stages of osteoarthritis.

It has been shown that in vitro treatment with Irisin on human chondrocytes, obtained from patients with osteoarthritis, increases the synthesis of glycosaminoglycans and type II collagen and reduces the expression of type X collagen and some inflammatory cytokines, such as interleukins 1 and 6 [14]. An in vivo study has shown that intra-articular administration of Irisin in a mouse model of osteoarthritis reduces cartilage erosion and inflammation of the synovial membrane [15].

The development of new molecules to prevent/treat changes in the articular cartilage could reduce the risk of developing osteoarthritis and the consequent health costs. Scientific research has long been seeking bioactive molecules by tissue engineering of cartilage with the aim of treating the early stages of osteoarthritis to prevent its long-term progression. A notable step forward was achieved by a study that was performed on murine cells of the ATDC5 chondrogenic cell line treated with 100 ng/mL of untagged recombinant Irisin for 24 h, in which it has been found an Irisin-mediated reduced expression of several inflammatory genes, including interleukin-1 beta, interleukin-6, and tumor necrosis factor-alpha. In addition, if chondrogenic lineage cells were treated for 7 and 14 days with Irisin, the expression of genes relevant to chondrocyte differentiation, such as Collagen type II and Aggrecan, were increased [16]. Furthermore, the study revealed that in primary chondrocytes that were isolated from Irisin knock-out mice, gene expression of cartilage anabolic marker genes was decreased, while the expression of inflammatory factors was increased, thus suggesting that Irisin is a pivotal factor for normal cartilage development and for the prevention of its degeneration [16].

However, making in vitro models that mimic intra-articular cartilage microarchitecture has been a very complex challenge, including whether there is any difference between murine and human cells. For this purpose, in the present study we performed biomolecular analysis and histomorphometry on three-dimensional cultures of human articular chondrocytes (HACs) that were treated with untagged recombinant Irisin, demonstrating that Irisin also exerts its anabolic potential on human chondrocytes from non-osteoarthritic (healthy) subjects. HACs in culture easily undergo the phenomenon of the de-differentiation, assuming a fibroblast-like morphology and, in monolayer culture, failing to produce cartilage matrix [17]; thus, we used three-dimensional cultures [18] to study the effect of Irisin on chondrocyte matrix deposition. Specifically, we show that the myokine exerts a receptor-mediated effect on these cells, as demonstrated by the rapid activation of ERK phosphorylation; up-regulates mRNA and protein levels of SRY-box transcription factor 9 (Sox-9) and Collagen type II; and finally, although it does not change the proteoglycans content, it enhances chondrocyte proliferation, as demonstrated by the increased pellet size of three-dimensional cultures of HACs, and the Collagen type II in the matrix.

## 2. Materials and Methods

### 2.1. Irisin Treatment

Irisin was sourced as an untagged recombinant protein that was produced in *E. coli*, previously validated by Elisa Assay [4], and purchased from Adipogen (Liestal, Switzerland), reconstituted at 100 µg/mL in milliQ water and used with a final concentration of 100 ng/mL (treatment group) in the experiments that were performed.

### 2.2. Chondrocyte Pellet Cultures

We used commercially available human chondrocytes (HCHON, Provitro^®^, Berlin, Germany). Provitro’s HACs cultures are derived from human original tissues (in vivo state). They are not transferred or mutated and have a limited in vitro lifespan. HACs were seeded at a cell density of 10 × 10^3^ cells/cm^2^ in D-MEM (Thermo Fisher Scientific, Waltham, MA, USA) with 10% heat inactivated fetal bovine serum (FBS), 1% penicillin/streptomycin (Thermo Fisher Scientific, Waltham, MA, USA), fibroblast growth factor 2 (FGF-2) 10 ng/mL, and transforming growth factor beta 1 (TGFβ_1_) 10 ng/mL (R&D systems) in monolayer culture until reaching 70–80% confluence. After 1 passage, the cells were trypsinized and placed in pellet cultures, as previously described [19], using polystyrene 15 mL tubes. Each tube containing 5 × 10^5^ cells was centrifuged at 500× *g* for 10 min at 4 °C. After several hours in the incubator, the pellets spontaneously assumed a perfectly spherical shape located at the bottom of the polystyrene tube. The spherical pellets were cultured in chondrogenic medium, supplemented with Irisin, only in the case of the Irisin-treated samples, added at every medium change, twice per week for 21 days. At the end of the culture time, the pellets from each group (Ctr and Irisin) were collected for histological analysis (6–9 pellets Ctr; 6–12 pellets Irisin-treated) and real-time PCR (5–6 pellets Ctr; 5–6 pellets Irisin-treated).

### 2.3. Real-Time PCR

The total RNA extraction was performed using a commercial kit (RNeasy, Qiagen, Hilden, Germany) and following the manufacturer’s instructions. Briefly, the samples were first lysed and then homogenized, by utilizing 𝛽-mercaptoethanol and a specific buffer that was included in the kit. Ethanol was added to the lysate to provide ideal binding conditions. Spin columns (RNeasy, Qiagen, Hilden, Germany) were used for purification of the extracted RNA. Briefly, the lysate was loaded onto the RNeasy silica membrane and subjected to a series of centrifugations. SuperScript First-Strand Synthesis System kit (Bio-Rad iScript Reverse Transcription Supermix) was used to convert the RNA that was obtained (2 μg) to cDNA by reverse transcription (RT). The synthesized cDNA, in the amount of 20 ng, was then subjected to quantitative PCR. A Bio-Rad CFX96 Real-Time System with the SYBR Green PCR method as reported by the manufacturer’s protocol (Bio-Rad Sso Fast Evagreen Super Mix) was utilized for real-time PCR analysis. The mean cycle threshold value (Ct) from triplicate samples was used to calculate gene expression, and cDNA was normalized to the average of β-actin and β_2_ microglobulin (B2M) levels for each reaction.

The following primer pairs were used for the RT-PCR amplification: sense Sox-9 5′-GCTCTGGAGACTTCTGAACGAGAG-3′;antisense Sox-9 5′-CGTTCTTCACCGACTTCCTCC-3′;sense Aggrecan 5′-GACTTCCGCTGGTCAGATGG-3′;antisense Aggrecan 5′-RCGTTTGTAGGTGGTGGCTGTG-3′;sense Collagen I 5′-TGAAGGGACACAGAGGTTTCAG-3′;antisense Collagen I 5′-GTAGCACCATCATTTCCACGA-3′;sense Collagen II 5′-CATGAGGGCGCGGTAGAGAC-3′;antisense Collagen II 5′-TGCCAGCCTCCTGGACATC-3′;sense Collagen X 5′-AACTCCCAGCACGCAGAATCC-3′;antisense Collagen X 5′- GGCATTTGGTATCGTTCAGCG-3′;sense β-actin 5′-AATCGTGCGTGACATTAAG-3′;antisense β-actin 5′-GAAGGAAGGCTGGAAGAG-3′;sense β_2_ microglobulin (B2M) 5′-ATGAGTATGCCTGCCGTGTGA-3′;antisense β_2_ microglobulin 5′-GGCATCTTCAAACCTCCATG-3′;

### 2.4. Western Blot

SDS-PAGE gel electrophoresis and Western blot analysis were performed to analyze the protein levels of ERK phosphorylation (pERK). HACs were starved for at least 3 h, in serum-free D-MEM and stimulated for 1, 5, 7, 10, or 20 min supplementing the medium with Irisin only in the case of the Irisin-treated samples. At each time point, the cells were lysed (3 Ctr; 3 Irisin-treated) as already described [20,21] and then the Bio-Rad protein assay was used to detect the total protein concentration of the lysates. Equal amounts of protein for each sample were separated by SDS-PAGE, transferred to nitrocellulose membranes (Invitrogen, Carlsbad, CA, USA), and then probed with primary and secondary antibodies. The Odyssey Infrared Imaging System of LI-COR (LI-COR Biotechnology, Lincoln, NE, USA) was used for the visualization of the immune complexes. Anti-pERK, anti-totalERK, and β-Actin antibodies were purchased from Santa Cruz Biotechnology.

### 2.5. Histological Analysis

Chondrocyte pellets were fixed with 4% PFA in PBS, then dehydrated through alcohol series and embedded in paraffin. The paraffin-embedded blocks were sectioned at 5 μm thickness on a standard microtome (RM-2155 Leica, Heidelberg, Germany). Subsequently 0.1% Safranin-O (Merck Millipore, Danvers, MA, USA) was carried out on deparaffinized sections to assess the proteoglycan content of the cartilage, which stained orange to red. Histological sections were observed and photographed under the optical microscope (Leica) making use of 10X, 20X, and 40X objective lenses, then analyzed with Image-J software (Research Services Branch, Image Analysis Software Version 1.53a, NIH, Bethesda, MD, USA).

### 2.6. Immunofluorescence of Pellet Sections

Pellet sections that were chosen for the immunofluorescence analysis were subjected to deparaffinization followed by hydration with ethanol (100% twice—95%—80%—70%) and distilled water using incubations of 10 min each. Subsequently, the antigen retrieval was performed with Pepsin in Tris-HCl (pH 2.0, 1M) at RT for 15 min. The sections were then blocked in 3% BSA, 0.5% Triton X in PBS for 1 h. Afterwards, the samples were incubated with Collagen type II (MAB8887, Sigma-Aldrich, St. Louis, MO, USA), Sox-9 (702016, Invitrogen, Waltham, MA, USA), and Aggrecan (MA3-16888, Invitrogen, Waltham, MA, USA) primary antibodies o.n. at 4 °C in a humid chamber. After incubation and washing, the bound antibodies were detected using 2 μg/mL of fluorescently labeled goat anti-mouse secondary antibody (Alexa Fluor 488, Invitrogen) and goat anti-rabbit secondary antibody (Alexa Fluor 488, Invitrogen), with an incubation time of 1 h at RT. The samples were embedded in Mowiol containing 0.1% (*v*/*v*) DAPI for an additional staining of the nucleus. The sections were then visualized and photographed using a multispectral confocal microscope Leica TCS SP5. The images were adjusted in brightness and color with ImageJ software (Research Services Branch, Image Analysis Software Version 1.53a, NIH, Bethesda, MD, USA).

### 2.7. Statistical Analysis

A Mann–Whitney U test was used to assess statistical significance (*p* < 0.05), since the sample size <30, using GraphPad Prism version 8.0.2 for Mac OS (GraphPad Software, La Jolla, CA, USA, www.graphpad.com accessed on 8 February 2023). The data are presented as boxplots with median and interquartile ranges in the case of figures with three values per group. For figures with more than four values per group, the data are displayed as box-and-whisker plots with median and interquartile ranges, from max to min. All the data points are shown. Images were processed with ImageJ software.

## 3. Results

### 3.1. Extracellular Signal-Regulated Kinase (ERK) Phosphorylation in HACs Treated with Irisin

HACs were cultured in vitro, in chondrogenic medium, subjected to starvation for at least 3 h, and subsequently stimulated with Irisin for different time points. The examination of ERK phosphorylation in whole cell lysates was performed by Western blot. As shown in Figure 1, Irisin stimulation significantly upregulated pERK expression after 10 min, and then returned to basal levels at 20 min. These results demonstrated that HACs are responsive to Irisin treatment by enhancing pERK, thus suggesting a direct receptor-mediated action of the myokine on human chondrocytes.

### 3.2. Irisin Treatment Increases Sox-9, Aggrecan, and Collagen II Expression in Chondrocyte Pellets

In our previous study [19], we observed that HACs, when grown in 2D cultures, tend to de-differentiate assuming a typical fibroblastic morphology. The cells were then amplified in vitro (Figure 2A) and subsequently cultured in 3D pellets (Figure 2B). The use of this strategy allows the mimicking of the three-dimensional micro-architecture of the tissue, preventing the de-differentiation of the chondrocytes that occurs very easily when cells are grown in two dimensions. The 3D pellet culture systems, known as the gold standard in the field of 3D cultures, have also been approved for clinical application for the purpose of restoring cartilage defects [22,23].

### 3.3. Effects of Irisin on the mRNA Expression of Chondrogenic Markers

In order to find out if Irisin could affect HACs chondrogenic differentiation, RT-PCR was performed on both groups of 3D pellets (Ctr and treatment) after 21 days of culture to evaluate the mRNA expression level of the typical markers of chondrogenic differentiation. Figure 3 shows that Sox-9, Aggrecan, and Collagen II mRNA levels were significantly increased in the pellet cultures that were treated with Irisin (Figure 3A–C). Notably, the mRNA expression level of Collagen I, which is expressed in de-differentiated chondrocytes, does not appear to be affected by the presence of the myokine (Figure 3D). Consequently, the ratio of mRNA levels of Collagen II to Collagen I (CII/ CI), which can be considered as a differentiation index of chondrocytes, was statistically higher in the treatment group compared to the control (Figure 3E). Furthermore, Collagen X mRNA expression did not change significantly between the Irisin-treated and control chondrocyte pellets (Figure 3F).

### 3.4. Irisin Stimulates Chondrocyte Differentiation and Tissue Growth

In parallel, the chondrocyte pellets, whether they were treated with the cytokine under study or not, after 21 days of differentiation were fixed, embedded in paraffin, sectioned, and subjected to histological analysis. Morphometric analysis of the pellet sections allowed us to quantify their dimensions and to obtain a theoretical reconstruction of their thickness.

At the end of the differentiation period, the pellets were cut into sections of 5 µm each and stained with Safranin to spotlight chondrocytes (Figure 4A,B). The use of this specific staining for the cartilage matrix revealed its presence, demonstrating the ability of the cells to differentiate and produce the components of the cartilage matrix in the culture conditions that were used. The intensity of Safranin-O (Red) staining is directly proportional to the proteoglycan content inside the pellet. The first observation we made was related to the number of sections that were obtained for each pellet analyzed: the treatment with Irisin obtained a greater number of sections than the control indicating a greater cell proliferation and/or matrix secretion. Using ImageJ software, we analyzed the Safranin staining of the individual sections and, as the graph shows (Figure 4C), there was no significant difference in the percentage of Safranin-stained pellet area between control and Irisin treatment. Interestingly, however, the area of the Irisin-treated pellet sections was significantly larger than that of the control pellets (Figure 4D).

### 3.5. Irisin Modulates HACs Chondrogenic Differentiation

Immunofluorescence analyses were used to identify the expression of typical chondrogenic markers, giving the possibility to verify whether the treatment with Irisin had an influence on chondrogenic differentiation and activity. Interestingly, the pellets that were exposed to Irisin treatment showed increased positivity for Sox-9 (Figure 5A,B) and Collagen II (Figure 5C,D), confirming the upregulation of their mRNA levels (Figure 3).

### 3.6. Irisin Does Not Affect the Expression of Aggrecan in HAC Pellets

Differently to the modulation of its gene expression (Figure 3B), the quantization of positivity for Aggrecan in HAC pellets, measured by immunofluorescence, was not changed by Irisin treatment (Figure 6A,B).

## 4. Discussion

Nowadays, although the increase in life expectancy is considered one of the main achievements of modern science, at the same time the increase in the median age of the population also leads to a higher incidence of chronic diseases in the elderly.

Over the past 30 years, there has been a 9.3% increase in the age-standardized prevalence and incidence rate of hip and knee osteoarthritis cases, suggesting that the burden of osteoarthritis will increase over time and may represent a major cause of years lived with disability, globally [24]. Since this disease is associated with advanced age and obesity, the increase of these two characteristics in the population will lead to significant increases in the prevalence of osteoarthritis. Unfavorable body composition contributes to the development of osteoporosis and osteoarthritis, but while a low BMI (body mass index) is associated with an increased risk of osteoporosis, obesity stimulates the development of osteoarthritis due to the increased mechanical load on the weight-bearing joints as well as for presumed unfavorable effects of specific adipokines of the adipose tissue involved in cartilage degradation [13]. If an elderly person develops osteoarthritis as a comorbidity of osteoporosis, the negative effects on bone and muscle quality are significantly greater. Since exercise, both strengthening and aerobic training, are known to improve joint function in patients suffering from osteoarthritis [25], several studies have investigated the possible protective role of the myokine Irisin in this pathology [14,16].

The identification of new therapeutic approaches for osteoarthritis would be useful for a disease that is characterized by the degeneration of cartilage and associated with aging in a large part of the population.

Our aim was to study Irisin ability to induce the proliferation, differentiation, and matrix secretion of human chondrocytes from healthy subjects and, therefore, to add a piece to the knowledge of the possible effects of this molecule on cartilage tissue.

The evaluation of chondrocyte responsiveness to Irisin treatment was carried out focusing on the signaling pathway to observe the phosphorylation of the MAP kinase ERK. Irisin functional receptor, in fact, is still a critical point of study. Although it has recently been shown that Irisin acts on osteocytes by binding an αV/β5 integrin receptor [26], there are still numerous unresolved questions regarding its mechanism of action. However, it is well known that stimulation of the ERK cascade is among the signaling events that are triggered by the binding of a ligand to its integrin receptor [27]; moreover, it has been demonstrated that Irisin induces osteoblastic proliferation and differentiation through pERK [28] and increases the expression of the transcription factor Atf4 through an Erk-dependent pathway in osteocytes [4]. Consistent with this result, our data showed that Irisin treatment stimulated receptor-mediated pERK activation, albeit later than in osteoblasts and osteocytes in which it occurs after 5 min [2,10]. Therefore, these data may support the conclusion that chondrocytes express the receptor for Irisin, possibly the αV/β5 integrin.

As we and others have previously observed, HACs grown in 2D cultures tend to de-differentiate by assuming a typical fibroblastic morphology [19]. A strength of the present study is that the cells were cultured in 3D pellets. This strategy allowed us to evaluate the effect of Irisin on chondrocytes by mimicking the three-dimensional microarchitecture of the tissue. Since 3D pellet culture systems have also been approved for clinical application in cartilage tissue engineering, our data, showing an Irisin-mediated increase in chondrocyte differentiation and Collagen II production, might support the potential use of Irisin for cartilage regeneration. In agreement with these findings, another study has recently emerged to support the relevance of Irisin for its chondroprotective action. A compelling study has shown that Irisin plays a role in promoting chondrogenic differentiation of human mesenchymal stem cells (hMSCs), which are currently being tested for articular cartilage repair in tissue engineering [29]. The authors showed that hMSCs, when cultured in the presence of Irisin in the chondrogenic differentiation medium for 7 and 14 days, exhibited higher levels than controls of Collagen type II, Aggrecan, and Sox-9, proving an Irisin-mediated effect through the activation of Rap1 and PI3K/AKT pathways [29]. Of note, similar to our results demonstrating significantly larger area in pellet sections of HACs that were treated with Irisin for 21 days compared with control pellets, Chen and colleagues demonstrated that cartilage tissues, induced from Irisin-treated hMSCs, were larger than controls at days 7 and 14 [29]. Moreover, we observed that Irisin treatment did not affect the expression of Collagen X, which is considered a specific marker of OA cartilage since it is highly expressed by hypertrophic chondrocytes [30]. This scenario could be explained considering that the strategy of using 3D pellets has been shown to strongly induce the expression of typical chondrogenic markers, while keeping the expression of hypertrophic marker genes relatively low [31]. Irisin probably does not have an evident effect on Collagen X as it already remains poorly expressed in the 3D system that was used.

Some limitations of our study include the analysis of chondrocyte pellets at a single time point (21 days), instead of a sequential study that would have been more interesting. Moreover, the results were obtained only through an in vitro experimental design, albeit on human primary cells. Very recently, the elegant in vivo study by Li et al. [16] showed that intra-articular injection with Irisin inhibits osteoarthritis progression in a surgically-induced pathological mouse model. They also found that Irisin knock-out mice developed more severe osteoarthritis than that of age-matched control mice while intra-articular injection of Irisin attenuated disease progression. However, for a more comprehensive understanding of the possible effects of Irisin on cartilage healing during aging, it would be desirable for further studies to decipher whether local or systemic treatment with the myokine in wild-type elderly mouse models could slow the progression of osteoarthritis.

Compared with the previous study that was performed in vitro on human chondrocytes that were isolated from patients with severe osteoarthritis [14], the present study provides a step forward by evaluating the effect of Irisin on HACs under physiological conditions, devoid of confounding factors such as osteoarthritis. In this regard, there is mounting evidence that chondrocytes in osteoarthritis share many of the typical changes of cellular senescence. In particular, a decline in the proliferative and anabolic response of chondrocytes to growth factor stimulation was observed [32].

Overall, these results, if confirmed in animal models, could have great application relevance for those patients who are unable to perform physical activity and consequently have an inadequate stimulus to support the production of Irisin. Therefore, this would allow the development of exercise-mimetic drugs that could be widely used in various cartilage pathological conditions.

## Figures and Tables

**Figure 1 pharmaceutics-15-00585-f001:**
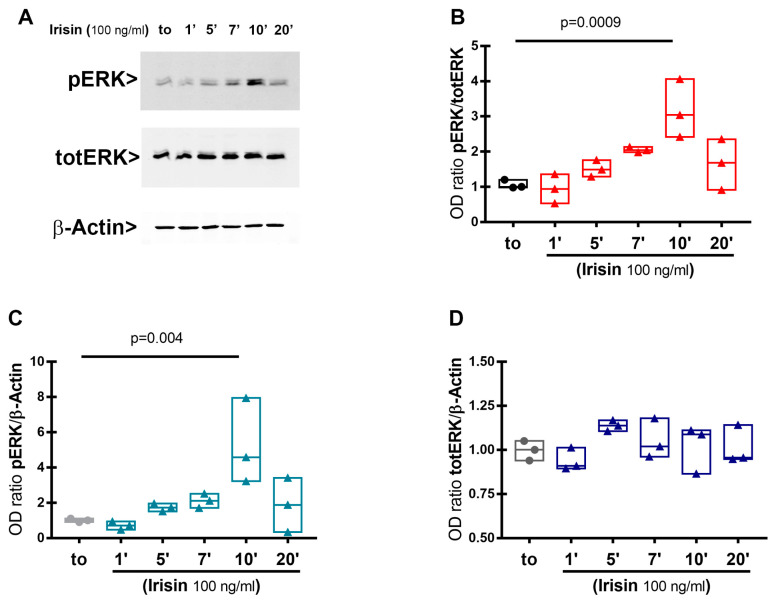
Irisin effect on ERK phosphorylation (pERK) in HACs cultures. HACs were treated with Irisin for the indicated time points. Representative image of phosphorylated ERK, Total ERK, and β-Actin levels in cell lysates analyzed by Western blotting (**A**). Expression levels of pERK, TotalERK, and β-Actin were measured with corresponding densitometric quantification. Densitometric analysis of the related bands was expressed as relative optical density of the bands, normalized to TotalERK (**B**) and β-Actin (**C**). Boxplot showing the normalization of TotalERK to β-Actin (**D**). Data are presented as boxplots with median and interquartile ranges.

**Figure 2 pharmaceutics-15-00585-f002:**
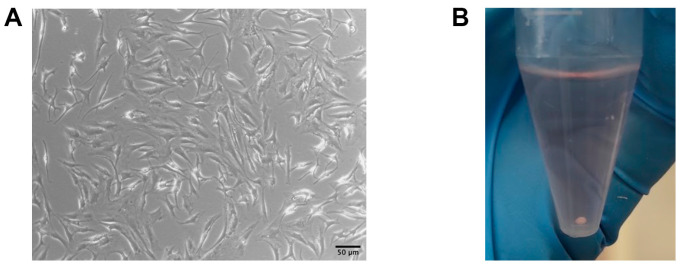
Two-dimensional and three-dimensional culture of HACs. A representative phase contrast picture of HACs cultivated in chondrogenic medium. Scale bar: 50 μm (**A**). An illustrative photo of a HAC 3D pellet cultured in a polystyrene 15 mL tube (**B**).

**Figure 3 pharmaceutics-15-00585-f003:**
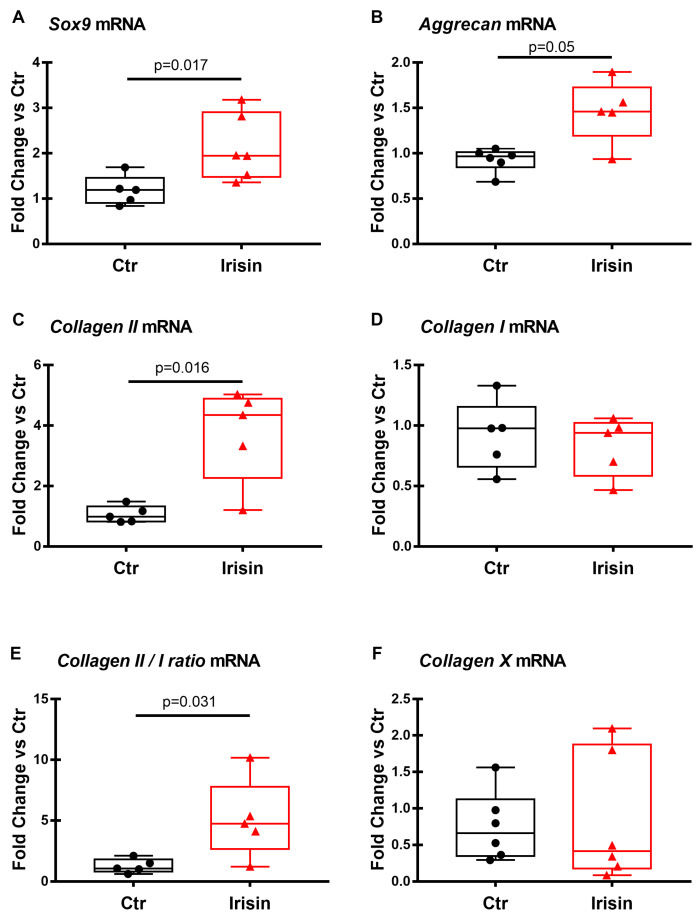
Effects of Irisin on the mRNA expression of chondrogenic markers. mRNA expression levels of chondrogenic marker genes Sox-9 (**A**), Aggrecan (**B**), Collagen II (**C**), Collagen I (**D**), and Collagen X (**F**) were analyzed (qPCR) after 21 days of stimulation with Irisin. The graphs show that the treatment significantly increased the expression of Sox -9, Aggrecan, Collagen II, and Collagen II / I ratio (**E**). Expression was normalized to the average of β-actin and B2M levels for each reaction. Data are shown as box-and-whisker plots with median and interquartile ranges, from max to min. A Mann–Whitney test was used to compare groups. *p*-values as indicated.

**Figure 4 pharmaceutics-15-00585-f004:**
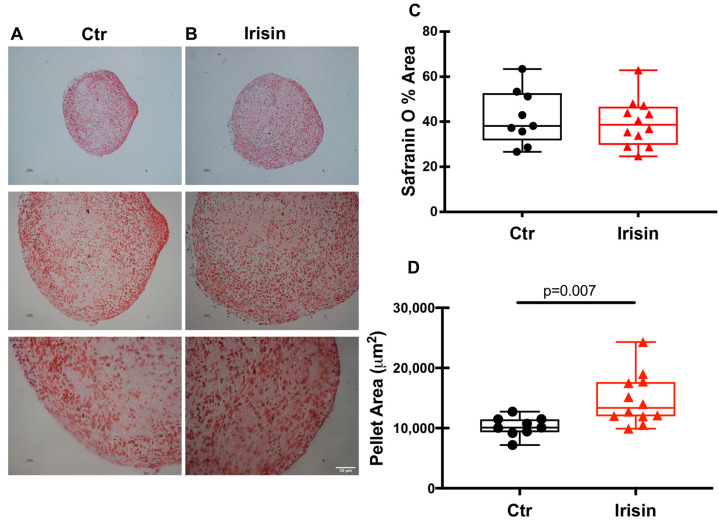
Safranin-O staining of HAC pellets in the presence or absence of Irisin. hACs in 3D culture were cultured for 21 days in control (Ctr) conditions (**A**) or in the presence of the myokine Irisin (**B**). Scale bar: 50 μm. Dot-plot graphs showing the quantification of % Safranin O area (**C**) and pellet area (**D**). Data are shown as box-and-whisker plots with median and interquartile ranges, from max. to min. A Mann–Whitney test was used to compare groups.

**Figure 5 pharmaceutics-15-00585-f005:**
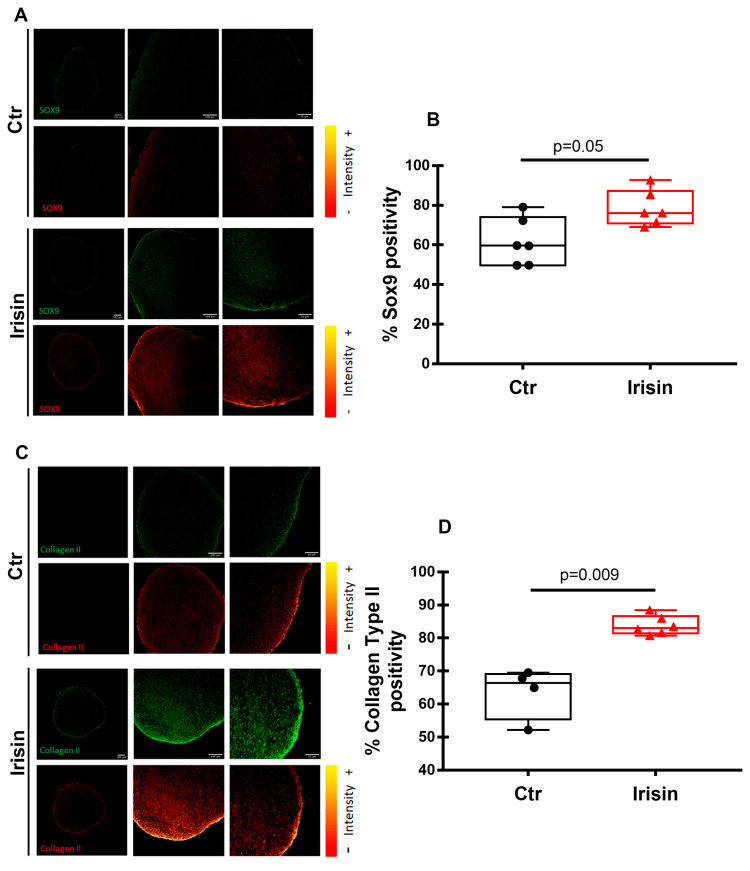
Immunofluorescence analysis of the chondrogenic markers in HAC pellets. Indirect immunofluorescence staining of Sox-9 (**A**) and Collagen II (**C**) in HAC pellets, treated with Irisin and Ctr, after 21 days of chondrogenic differentiation. For each panel, upper row: pellets stained for Sox-9 and Collagen II, green color represents the positive reaction. Lower row: lookup table displaying the localization of Sox-9 and Collagen II, the color bar on the right reflects the range of pixel intensities from red (lower) to yellow (higher). The figures are representative of four to six separate experiments. Dot-plot graphs showing the quantification of % positivity of Sox-9 (**B**) and Collagen II (**D**). Data are shown as box-and-whisker plots with median and interquartile ranges, from max to min. A Mann–Whitney test was used to compare the groups.

**Figure 6 pharmaceutics-15-00585-f006:**
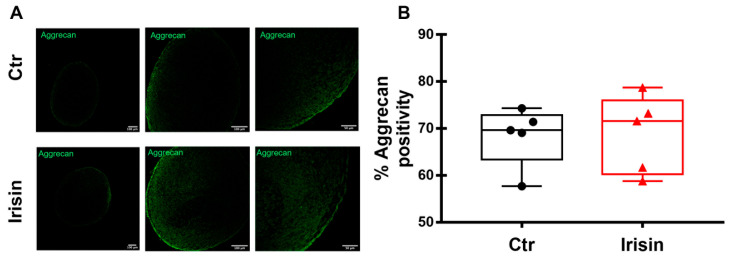
Immunofluorescence staining of Aggrecan (**A**) in HAC pellets, treated with Irisin and Ctr, after 21 days of chondrogenic differentiation. The figures are representative of 5 separate experiments. Dot-plot graphs showing the quantification of % positivity of Aggrecan (**B**). Data are shown as box-and-whisker plots with median and interquartile ranges, from max to min. A Mann–Whitney test was used to compare the groups.

## Data Availability

Data are available in a publicly accessible repository that does not issue DOIs. This data can be found here: [Data set Manuscript Posa et al.].

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
