# Peer review of "Irisin Role in Chondrocyte 3D Culture Differentiation and Its Possible Applications"

_pharmaceutics, 2023, doi:10.3390/pharmaceutics15020585_

Round 1

Reviewer 1 Report

The authors investigated the effect of irisin on re-differentiation of commercially available human chondrocytes in pellet culture. The study reproduces the results of the previous study (ref 15), which was performed using chondrocytes of patients with OA, therefore, the importance of the study is doubtful.

Comments

1.      The authors should describe the novelty and rationale for their study.

2.      Introduction should include description of every issue investigated in the study including signaling pathways.

3.      Line 28: The origin of human articular chondrocytes should be indicated here.

4.      Line 63: Term senolytic is not compatible with irisin as the authors state that it might serve as agent for cartilage regeneration. This should be corrected.

5.      Line 94: The irisin concentration used in the study (100ng/ml) should be justified as in the previous study (ref 15) another concentration was effective. The authors should demonstrate the effect of different concentrations of irisin in the chondrocytes used in the study.

6.      Lines 95, 108: Here the number of samples in each experimental group should be indicated.

7.      Line 112: Spin columns are used for purification of the extracted RNA. The method of RNA extraction should be also indicated.

8.      All the abbreviations should be disclosed at first use.

9.      Lines 150-152: Safranin-O stains proteoglycans in the cartilage, while Fast-Green staining is used for collagen. This should be corrected.

10.  Lines 156-167: Immunofluorescence protocol should be described in more detail.

11.  Lines 169-176: If the number of samples is less than 30, Mann-Whitney U-test should be always used as well as graphically box-and-whisker presentation should be applied. This should be corrected in all the graphical material.

12.  Fig 1A: Reference gene expression such as β-actin etc. should be included.

13.  Line 202: Reference is required at the end of this sentence.

14.  Line 211: Fig 2 does not contain this information. This should be corrected.

15.  Lines 208-217: If the authors aim to describe chondrocyte differentiation Type X collagen expression data should be also included.

16.  Line 323: The effect of irisin in mouse models has been already demonstrated (Front Cell Dev Biol 2021, 9:703670 Xiangfen Li et al).

17.  Lines 327-328: If the authors suggest using irisin for cartilage regeneration, it is not clear why conditions of OA inflammation are considered confounding. This should be clarified.

Author Response

The authors investigated the effect of irisin on re-differentiation of commercially available human chondrocytes in pellet culture. The study reproduces the results of the previous study (ref 15), which was performed using chondrocytes of patients with OA, therefore, the importance of the study is doubtful.

We thank the reviewer for thorough review of our manuscript and constructive criticism. We are grateful for the very useful comments that helped us improving the quality of our paper. Below is a rebuttal letter with a point-by-point response to each of the reviewer's comments.

Comments

  1. The authors should describe the novelty and rationale for their study
  2. Introduction should include description of every issue investigated in the study including signaling pathways.

Welcoming the Reviewer's indication regarding points 1 and 2, we have rephrased the final part of the Introduction, describing the rationale and novelty of our study, as well as we have added a summary with the relevant results of our study.

  1. Line 28: The origin of human articular chondrocytes should be indicated here.

As appropriately requested by Reviewer, we added in the Abstract that we used commercially available human articular chondrocytes. We also specified in Material and Method section that Human chondrocytes (HACs) purchased from Provitro® are derived from human original tissues (in vivo state) and not transferred or mutated and have a limited in vitro lifespan.

  1. Line 63: Term senolytic is not compatible with irisin as the authors state that it might serve as agent for cartilage regeneration. This should be corrected.

As per Reviewer's suggestion, we corrected the sentence.

  1. Line 94: The irisin concentration used in the study (100ng/ml) should be justified as in the previous study (ref 15) another concentration was effective. The authors should demonstrate the effect of different concentrations of irisin in the chondrocytes used in the study.

In response to the reviewer's comment, we would like to specify that the type of irisin used in this study is different from that used by Vadalà et al (https://doi.org/10.3390/cells9061478). Briefly, we used untagged recombinant protein produced in E. coli (Adipogen, Liestal, Switzerland), previously validated by Elisa Assay (DOI: 10.1002/jbmr.3944), better described now in the revised Materials and Methods section. Vadalà et al. used r-irisin (Sigma, St. Louis, MO, USA) which is a recombinant protein, expressed in CHO cells, FLAG tagged at the N-terminus.

Nevertheless, we considered as the starting point of our study to assess whether irisin activated a receptor-mediated signal in chondrocytes, showing that the effective dose to activate MAP kinase ERK phosphorylation was 100 ng/ml, as already shown in previous studies (doi: 10.1073/pnas.1516622112; doi: 10.1002/jbmr.3944; doi: 10.1002/jbmr.4192). Conversely, Vadalà et al. observed that p-ERK protein levels did not show significant changes in hOAC treated with 25 ng/ml of His-tag N-terminal irisin.

  1. Lines 95, 108: Here the number of samples in each experimental group should be indicated.

In response to the Reviewer's suggestion, we indicated the number of samples in each experimental group on lines 117-120 of the revised Manuscript (paragraph: Chondrocyte Pellet Cultures). In addition, we performed new experiments on chondrocytes pellets to increase the sample size for gene expression analysis by Real-Time PCR.

  1. Line 112: Spin columns are used for purification of the extracted RNA. The method of RNA extraction should be also indicated.

We thank the reviewer for picking this up. We modified the paragraph Real-Time PCR in agreement with his/her suggestion.

  1. All the abbreviations should be disclosed at first use.

We thank the reviewer for the observation, we have checked all the manuscript accordingly.

  1. Lines 150-152: Safranin-O stains proteoglycans in the cartilage, while Fast-Green staining is used for collagen. This should be corrected.

We thank the reviewer for picking this up. We modified the sentence accordingly.

  1. Lines 156-167: Immunofluorescence protocol should be described in more detail.

In response to the Reviewer's suggestion, the paragraph on Immunofluorescence of pellet sections has been described in more detail.

  1. Lines 169-176: If the number of samples is less than 30, Mann-Whitney U-test should be always used as well as graphically box-and-whisker presentation should be applied. This should be corrected in all the graphical material.

In agreement with the Reviewer’s comment, we modified all graphs accordingly and changed the description of the Statistical Analysis paragraph as follow: because of sample size <30, significance was evaluated with Mann Whitney U test using GraphPad Prism version 7.0d for Mac OS X (GraphPad Software, La Jolla California USA, www. graphpad.com). For figures with three values per group, data are presented as boxplot with median and interquartile ranges. For figures with more than three values per group, data are presented as box-and-whisker plots with median and interquartile ranges, from max to min. All data points are shown.

  1. Fig 1A: Reference gene expression such as β-actin etc. should be included.

In response to Reviewer’s suggestion, we believe that benefit of looking at both protein forms, phosphorylated (pERK) and total (TotalERK) simultaneously, is the ability to control the variability with normalization. In this manner, the signal from the phosphorylated protein was normalized to the signal from the total amount of protein present in the cell lysates. In our case, TotalERK expression can be considered an internal control of pERK amount, as well as a control loading reference which is an adequate substitute of β-actin.

  1. Line 202: Reference is required at the end of this sentence.

As appropriately requested by Reviewer, References have been included at the end of the sentence.

  1. Line 211: Fig 2 does not contain this information. This should be corrected.

We thank the reviewer for picking this up. We corrected the typo.

  1. Lines 208-217: If the authors aim to describe chondrocyte differentiation Type X collagen expression data should be also included.

Agreeing to the reviewer's suggestion, we analyzed the expression Type X collagen. However, we found no significant differences between control- and Irisin-treated pellets. Results have been included in the revised Figure 3.

  1. Line 323: The effect of irisin in mouse models has been already demonstrated (Front Cell Dev Biol 2021, 9:703670 Xiangfen Li et al).

In response to the reviewer's comment, we deleted the sentence and properly mentioned the findings of Li X. and colleagues in the Introduction and the Discussion.

  1. Lines 327-328: If the authors suggest using irisin for cartilage regeneration, it is not clear why conditions of OA inflammation are considered confounding. This should be clarified.

In agreement with the reviewer’s suggestion, we have rephrased the sentence to clarify why we investigated the effects of Irisin in chondrocytes under physiological conditions.

Reviewer 2 Report

Interesting, well-designed work on irisin chondrogenicity.

Abstract OK

Intro: OK

Methods: it could have been interesting to compare the effect of irisin on chondrogenic redifferentiation and irisin chondrogenicity on MSCs (DOI: 10.1186/s13287-022-03092-8).

A sequential study could have been more interesting.

PCR Coll X?

Results: l211 : Figure 3

For each PCR the fold is rather Low (an internal control like TGF-ß1 could have been interesting).

Alizarin red staining and Aggrecan immunostaining?

Alcian Blue and osteogenic stanings?

The lack of difference between 4a & 4B in terms of staining is surprising.

Discussion OK and ref OK (please add DOI: 10.1186/s13287-022-03092-8)

Author Response

Interesting, well-designed work on irisin chondrogenicity.

We thank the reviewer for his/her feedback and constructive comments that helped in our efforts to improve the manuscript.

Abstract OK

Intro: OK

Methods: it could have been interesting to compare the effect of irisin on chondrogenic redifferentiation and irisin chondrogenicity on MSCs (DOI: 10.1186/s13287-022-03092-8).

We greatly appreciated the Reviewer's recommendation in mentioning this important recent study on another cell type. We added in the Discussion the similarity of the results on human mesenchymal stem cells and human articular cartilage cells regarding chondrogenicity of irisin, although analyzed at different time points.

A sequential study could have been more interesting.

As appropriately requested by the Reviewer, we added in the Discussion that some limitations of our study include the analysis of chondrocyte pellets at a single time point (21 days), instead of a sequential study that would have been more interesting.

PCR Coll X?

Agreeing to the reviewer's suggestion, we analyzed the expression Type X collagen. However, we found no significant differences between control- and Irisin-treated pellets. Results have been included in the revised Figure 3.

Results: l211 : Figure 3

We thank you the Reviewer for picking this up. There was a typo on line 211, mismatching the figure number. We have corrected the text substituting “Fig.2” with “Fig.3”.

For each PCR the fold is rather Low (an internal control like TGF-ß1 could have been interesting).

In response to the Reviewer's suggestion, we performed new experiments on chondrocytes pellets to increase the sample size for gene expression analysis by Real-Time PCR.

Alizarin red staining and Aggrecan immunostaining? Alcian Blue and osteogenic stanings?

In agreement with the Reviewer’s suggestion, we performed Aggrecan immunostaining. However, in contrast to modulation of its gene expression (Figure 3B), the quantization of positivity for Aggrecan in HAC pellets, measured by immunofluorescence, was not changed by Irisin treatment (new result on Figure 6). Regarding the request for other staining, unfortunately we dedicated all the pellets scheduled for histological analysis to Safranin O staining, preferring for Revision to use the spare pellets to perform immunofluorescence for Aggrecan.

The lack of difference between 4a & 4B in terms of staining is surprising.

With reference to the results described in Figure 4, we believe that, although Irisin does not change the proteoglycans content, it enhances chondrocyte proliferation, and the amount of Collagen type II in the matrix. We added this explanation in the last sentence of the Introduction.

Discussion OK and ref OK (please add DOI: 10.1186/s13287-022-03092-8)

We have mentioned this important work as suggested by the reviewer.

Reviewer 3 Report

The Authors present a paper entitled “Irisin role in chondrocyte 3D culture differentiation and its

possible applications” in which 3D cultures of human articular chondrocytes (HACs) have been treated with Irisin, showing a new crosstalk mechanisms between muscle and cartilage tissue. Furthermore, the confirmation of Irisin ability to induce chondrogenic differentiation could favor the development of exercise-mimetic drugs, with application relevance for patients who cannot perform physical activity. The paper is globally well written with sufficient background and methodological details. 

The work is interesting, nevertheless the Authors should address several points in order to improve the paper.

  • Authors should justify the decrease of pERK at time 20’ after Irisin stimulation. Are other molecular actors involved in the modulation along with pERK? 
  • The Authors should provide images of the 3D cultures, as they have provided for 2D
  • Moreover, they must justify why chondrocytes are a good model to monitor differentiation induced by Irisin, as I would suggest to use adipose derived stem cells to this purpose, please justify
  • Provide explanation for the difference in the area between ctr group and those treated with Irisin 
  • The discussion should be shortened especially in the first part of epidemiological analysis 

Author Response

The Authors present a paper entitled “Irisin role in chondrocyte 3D culture differentiation and its

possible applications” in which 3D cultures of human articular chondrocytes (HACs) have been treated with Irisin, showing a new crosstalk mechanisms between muscle and cartilage tissue. Furthermore, the confirmation of Irisin ability to induce chondrogenic differentiation could favor the development of exercise-mimetic drugs, with application relevance for patients who cannot perform physical activity. The paper is globally well written with sufficient background and methodological details. 

We really appreciate the reviewer’s comments and suggestions and we have tried our best to meet them all.

The work is interesting, nevertheless the Authors should address several points in order to improve the paper.

  • Authors should justify the decrease of pERK at time 20’ after Irisin stimulation. Are other molecular actors involved in the modulation along with pERK? 

We thank the reviewer for this thoughtful input. As showed in Figure 1, we found an activation of the signal, namely phosphorylation of ERK, that peaks at 10 minutes and then returns to basal levels at 20 minutes. In the description of Result of Figure 1, we have better specified that phosphorylation of ERK, which is normally transient, begins to decrease at 20 minutes.

Although we cannot demonstrate whether there are other molecular mediators upstream of ERK phosphorylation, we believe that, as observed in FGF2-stimulated chondrocytes (https://doi.org/10.1073/pnas.97.3.1113), the activation of pERK, the last molecular actor going into the nucleus, is likely responsible for up-regulation of the transcription factor Sox9.

  • The Authors should provide images of the 3D cultures, as they have provided for 2D

In response to the reviewer's request, we would like to specify that in Figure 2B we have included a representative photo of a 3D chondrocyte pellet cultured in a 15 mL polystyrene tube. As we have previously described (https://doi.org/10.3389/fendo.2020.00285), after passage in monolayer cultures, the cells were trypsinized, counted and divided into 15-mL polystyrene tubes. Each tube containing 5×105 cells was centrifuged at 500 g for 10 min at 4°C. After several hours in the incubator, the pellets spontaneously assumed a perfectly spherical shape located at the bottom of the polystyrene tube. We have included a more detailed description in Material and Methods, section Chondrocyte Pellet Cultures.

  • Moreover, they must justify why chondrocytes are a good model to monitor differentiation induced by Irisin, as I would suggest to use adipose derived stem cells to this purpose, please justify

In relation to this important point raised by the Reviewer, we would like to specify that previous studies have already demonstrated the influence of Irisin on the cellular differentiation of several other cell types, such as bone marrow mesenchymal cells (doi: 10.1073/pnas.1516622112) or, more recently, mesenchymal stem cells from dental tissue (doi: 10.3390/biology10040295). However, as rightly pointed out by the Reviewer, the chondrocyte is a type of cell already in a differentiation state. Therefore, the purpose of the present work, was not to analyze differentiation per se, but to assess whether Irisin was able to increase the activity of healthy human chondrocytes (e.g., Collagen II production) by modulating key transcription factors of differentiation (e.g., Sox9). We better detailed our aim in the final paragraph of the Introduction.

  • Provide explanation for the difference in the area between ctr group and those treated with Irisin 

In agreement with the reviewer’s suggestion, we provided explanation for the higher pellet area of Irisin-treated chondrocytes than controls. With reference to the results described in Figure 4, we believe that, although Irisin does not change the proteoglycans content, it enhances chondrocyte proliferation, and the amount of Collagen type II in the matrix.

  • The discussion should be shortened especially in the first part of epidemiological analysis 

We have shortened the Discussion as suggested by the Reviewer.

Round 2

Reviewer 1 Report

The authors improved presentation of their results, however, some questions are remained.

Comments

1.      The authors did not address the main concern of the previous review related to the rationale of investigation of Irisin effect on healthy chondrocytes. Healthy chondrocytes are terminally differentiated cells. On the other hand, chondrocyte differentiation is observed in the course of OA development. If Irisin induces differentiation of healthy chondrocytes, is it arthritogenic? This should be clarified.

2.      Introduction should describe every issue that was further examinerd in the study including for example, ERK signaling. This should be corrected.

3.      Line 97: ATDC5 is a correct writing.

4.      Line 130-149: It is not clear when Irisin was added to the culture and how long chondrocytes were cultured in its presence. This should be clarified.

5.      Lines 166-178: Collagen  type X primers should be included.

6.      Lines 180-190;232-239: If no housekeeping protein was used, the authors should measure exact amounts of totalERK concentrations in each of the examined samples and should prove that these amounts are constant.

7.      Lines 272-273: The absence of significant changes in Collagen type X  gene expression in response to Irisin treatments should be discussed in the Discussion section.

8.      Line 294: No evidence of cell differentiation was presented.  This should be clarified.

9.      Line 384: No receptor for Irisin was examined in the study. This should be corrected.

10.  Lines 385-386: This sentence is not clear. This should be clarified.

Author Response

Reviewer 1

The authors improved presentation of their results, however, some questions are remained.

Comments

  1. The authors did not address the main concern of the previous review related to the rationale of investigation of Irisin effect on healthy chondrocytes. Healthy chondrocytes are terminally differentiated cells. On the other hand, chondrocyte differentiation is observed in the course of OA development. If Irisin induces differentiation of healthy chondrocytes, is it arthritogenic? This should be clarified

We thank the reviewer for the remark: although the previous clarification, this issue could still be misunderstood. Irisin prompted chondrocyte differentiation increasing Collagen II secretion and Sox-9 expression which mimic the behavior of chondroblast during physiological cartilage matrix formation. In our experiments we found an increase of Collagen II production which is not the protein secreted by the hypertrophic chondrocytes during OA development.  We have added two references about this issue.

  1. Introduction should describe every issue that was further examinerd in the study including for example, ERK signaling. This should be corrected.

We have added the information on ERK signaling in the final part of the introduction (lines 109-110) as requested by the reviewer.

  1. Line 97: ATDC5 is a correct writing.

We thank the reviewer for noticing the typo. We have corrected it.

  1. Line 130-149: It is not clear when Irisin was added to the culture and how long chondrocytes were cultured in its presence. This should be clarified.

We thank the reviewer for the observation. We have added this piece of information in the Materials and Methods Section, Chondrocyte Pellet Cultures paragraph.

  1. Lines 166-178: Collagen type X primers should be included.

We have inserted Collagen Type X primers in the primer list. Thank you.

  1. Lines 180-190;232-239: If no housekeeping protein was used, the authors should measure exact amounts of totalERK concentrations in each of the examined samples and should prove that these amounts are constant.

Following the suggestion of the reviewer, we have analyzed β-Actin protein expression levels in our samples and have revised the Fig.1 accordingly. Expression levels of pERK have been normalized also to β-Actin showing a trend comparable to the one obtained with normalization to TotalERK.

  1. Lines 272-273: The absence of significant changes in Collagen type X gene expression in response to Irisin treatments should be discussed in the Discussion section.

We thank the reviewer for pointing up this aspect and we apologize for the missing discussion. We have inserted a short consideration on Collagen X results in the Discussion section (Lines 608-614) and cited references #30 and #31 referring to studies regarding hypertrophic chondrocyte markers.

  1. Line 294: No evidence of cell differentiation was presented. This should be clarified.

      We thank the reviewer for asking to clarify the point regarding chondrocyte differentiation. We would like to point out that chondrocytes, when cultured in 2D, tend to de-differentiate losing their features. The 3D pellet culture strategy allows to create chondroblast natural niche, obtaining a chondrocyte re-differentiation: cells reappear to express the typical chondrogenic markers (Coll II, Aggrecan and SOX9). Irisin seems to be able to intensify the action of the 3D culture system by further inducing the expression of these markers. In this view, the myokine influences chondrocyte differentiation, often also referred to as re-differentiation.

  1. Line 384: No receptor for Irisin was examined in the study. This should be corrected.
  2. Lines 385-386: This sentence is not clear. This should be clarified.

To fulfil the last two requests of the reviewer, we have rephrased the paragraph in the Discussion section (lines 587-589) trying to clarify the conclusions that can be drawn from the results obtained with our study.

We asked a native speaker to help us with the style of our manuscript and hope that the manuscript is now easier to read.

Reviewer 3 Report

The Authors have replied to all the comments raised by the Reviewer.

Author Response

We thank the reviewer for his/her feedback and constructive comments that helped in our efforts to improve the manuscript.

Round 3

Reviewer 1 Report

I have no more comments. The study is very important and is well done now. Congratulations! Accept as is.